# The Role of Autophagy in the Female Reproduction System: For Beginners to Experts in This Field

**DOI:** 10.3390/biology12030373

**Published:** 2023-02-26

**Authors:** Akitoshi Nakashima, Atsushi Furuta, Kiyotaka Yamada, Mihoko Yoshida-Kawaguchi, Akemi Yamaki-Ushijima, Ippei Yasuda, Masami Ito, Satoshi Yamashita, Sayaka Tsuda, Satoshi Yoneda, Shibin Cheng, Surendra Sharma, Tomoko Shima

**Affiliations:** 1Department of Obstetrics and Gynecology, Toyama Autophagy Team in Gynecology and Obstetrics, University of Toyama, Toyama 930-0194, Japan; 2Departments of Pediatrics, Women and Infants Hospital of Rhode Island, Warren Alpert Medical School of Brown University, Providence, RI 02905, USA

**Keywords:** autophagy, autophagy flux, decidualization, ER stress, LC3, NBR1, oxidative stress, p62, syncytiotrophoblasts, TFEB

## Abstract

**Simple Summary:**

Autophagy, a fundamental mechanism for maintaining homeostasis, has been studied in the field of reproductive health. Accurate evaluation of autophagy activity is possible in cells and experimental animals, but it is difficult for fixed tissues, including human samples. In this review, we introduce some studies related to autophagy in this field and provide a tip for interpreting experimental results and understanding autophagy.

**Abstract:**

Autophagy is a fundamental process involved in regulating cellular homeostasis. Autophagy has been classically discovered as a cellular process that degrades cytoplasmic components non-selectively to produce energy. Over the past few decades, this process has been shown to work in energy production, as well as in the reduction of excessive proteins, damaged organelles, and membrane trafficking. It contributes to many human diseases, such as neurodegenerative diseases, carcinogenesis, diabetes mellitus, development, longevity, and reproduction. In this review, we provide important information for interpreting results related to autophagic experiments and present the role of autophagy in this field.

## 1. Introduction

Autophagy is a fundamental cellular process that is involved in regulating cellular homeostasis. This process is highly conserved from yeast to mammals and is always active at basal levels to recycle long-lived proteins, damaged organelles, and misfolded proteins. Some physiological or chemical stimuli, such as starvation, hypoxia, or rapamycin treatment, initiate autophagosome formation in the cytoplasm, which is a unique double-membrane structure. The autophagosome then fuses with lysosomes that contain acidic hydrolases, resulting in the degradation of the autophagosome content, including the inner but not the outer membrane. Eventually, the degraded proteins are reused to build new organelles, including lysosomes [1].

The number of autophagy-related studies has steeply increased in the scientific field, suggesting a correlation between autophagy and human diseases, including neurodegenerative diseases, cancers, and metabolic diseases. The number of studies on the role of autophagy in pregnancy, ovarian functions, gynecologic cancers, endometriosis, infertility, and sexually transmitted diseases has increased recently [2,3,4,5]. When we searched for papers related to “Autophagy and Reproduction” in PubMed, less than 50 papers were discovered until 2011. However, over 100 papers were published from 2013 every year, with more than 200 published since 2017. A reason for this increase could be the Nobel Prize in Physiology or Medicine awarded in 2016 to a Japanese scientist, Yoshinori Ohsumi, for the investigation of “autophagy” [6]. As this novel mechanism contributes to the prevention of some diseases, manipulation of autophagy in vivo could be a potential remedy. Although medicines that can modulate autophagic activity are not clinically available for some diseases, several investigators have attempted to develop novel medications to treat patients via autophagy modulation. In this paper, we first mention caution in autophagic experiments and introduce and review the role of autophagy in reproduction. 

## 2. Things That You Have to Know for Autophagic Experiments

This is the most crucial aspect of this manuscript. Please do not skip this and allow me to write this part in the first-person singular form.

I have been investigating autophagy in trophoblasts for more than 20 years. During these 20 years, methods for evaluating autophagy have been well developed, but observing autophagic activity in the human body remains difficult. Figure 1A shows the diagrammatic autophagic flow. The microtubule-associated protein 1 light chain 3 (LC3), a ubiquitin-like protein, is involved in autophagosome formation. LC3-I is produced from pro-LC3 following cleavage by the cysteine protease, autophagy-related protein 4 (ATG4). Upon autophagy activation, ATG4 converts LC3-I to the active form, LC3-II. Furthermore, p62 (also known as sequestosome 1, SQSTM1) accumulates and binds to the autophagosomes in cells with active autophagy. Although LC3-I is distributed diffusely throughout the cytoplasm, the accumulated LC3-II and p62 are present on the membrane and can be visualized as dots by immunocytochemistry. Both LC3-II and p62 are attached to the inner membrane of the autophagosome and are degraded together with the autophagosome content.

The first point of autophagic experiments is to understand that “Autophagy is a flow in cells for managing cellular proteins” (Figure 1B). Water circulates on Earth in different states, such as ice, liquid, and gas; water corresponds to proteins, earth to cells, and circulation to autophagy. Some readers who perform autophagic experiments would know that an autophagy inhibitor, bafilomycin A1 (BAF) or chloroquine, is used to evaluate autophagy flux, which indicates autophagic activity. Both BAF and chloroquine inhibit autophagy by suppressing autolysosomes but not autophagosomes. BAF decreases the acidity of lysosomes, thus inhibiting degradation within autolysosomes. Chloroquine blocks the lysosome-autophagosome fusion without affecting lysosomal acidity [7]. Second, it is important to know how to use autophagic inhibitors. To evaluate autophagy flux, we compared the ratio of LC3-II/actin and GAPDH in the presence and absence of BAF. The LC3-II/actin ratio is widely accepted as a parameter for evaluating autophagy; however, it can be replaced with the LC3-II/LC3-I ratio. BAF helped us determine the flow speed of autophagy via lysosomal inhibition (Figure 1C). In the absence of BAF, LC3-II protein is not detected in cells (Figure 1C, a and b); the accumulation of LC3-II protein is detected in the presence of BAF (Figure 1C, c and d). Thus, BAF exposure can visualize autophagy flux in cells by blocking the flow. Therefore, using this type of inhibitor is similar to creating a dam on a river. When the speed of the river is high, more water is pooled in the dam; meanwhile, less water is pooled when the speed is low. In this study, the amount of pooled water corresponds to the amount of LC3-II, a reliable marker of autophagy. 

After understanding autophagy flux, western blots for LC3 and p62 proteins can be performed to evaluate autophagy. It should be noted that any treatment that potentially activates autophagy does not always increase the LC3-II/actin ratio in the absence of BAF (Figure 1C and Figure 2A). Thus, to evaluate autophagy flux, BAF should be used. Conversely, treatments that increase the LC3-II/actin ratio in the presence of BAF might activate autophagy (Figure 2A, Tre-A). A concomitant reduction in the expression of p62, an autophagosome receptor protein, also suggests autophagy activation (Figure 2B, pattern A). However, a significant increase in p62 expression strongly suggests autophagy inhibition (Figure 2B, pattern B). Occasionally, no differences in p62 expression have been observed when autophagy is activated or inhibited (Figure 2B, pattern C). This phenomenon is explained in this study [8]. In addition, I have introduced some typical misleading results regarding autophagy. To show an increase in LC3 mRNA or protein without BAF and an increase in the mRNA levels of other autophagy-related molecules’ mRNA are used to demonstrate autophagic activation. In the absence of BAF, the increase in LC3-II protein levels is not evident because BAF does not increase LC3-II levels in cells with the blockade of autolysosome formation (Figure 2C, Pattern E). Thus, autophagy flux must increase when autophagy is activated (Figure 2C, Pattern D). From this perspective, it is impossible to demonstrate autophagy flux in the fixed tissues of humans. The increase in LC3 dots, which indicates LC3-II, in cells of the fixed organ, is construable in two-faced conclusions. One is the activation of autophagy in organs, and the other is the inhibition of autophagy during autolysosome formation. This is because an increase in LC3-II levels can occur in cells with lysosomal impairment. We reported that this type of autophagic disorder occurs in preeclamptic placentas [9]. An increase in autolysosomes observed using an electron microscope or other autophagy-related molecules might support the activation of autophagy; an increase in p62 or phosphorylated p62 might support autophagic inhibition. In fixed human tissues, there is no precise method to determine the status of autophagy. Therefore, these data should be interpreted with caution by integrating the in vitro and in vivo observations, clinical data, and our current understanding. In the future, studies should focus on identifying specific post-translational modifications (e.g., phosphorylation, acetylation, and O-GlcNAcylation) on ATG proteins or specific *ATG* mutations that can influence autophagic activity [10]. In mice, deletion of *autophagy-linked FYVE* (*Alfy*), which is required for brain development, results in the death of the pups within several hours after birth because of severe brain abnormalities [11]. Alfy mediates the elimination of ubiquitinated proteins, aggresome-like induced structures, and canonical aggregation-prone polyQ proteins, although it does not influence basal macroautophagy. Decreased Alfy levels and increased levels of aggregated proteins in fixed brain tissues suggest autophagy inhibition. Therefore, similar to *Alfy* in the brain, specific markers should be investigated for the female tract. At the cellular level, a quantitative method has been developed using degradable and non-degradable LC3 expression vectors [12]. In mice, conditional knockout (cKO) of *Atg7*, *Atg16L1*, *beclin-1 (BECN1)*, or *focal adhesion kinase family interacting protein of 200 kDa* (*FIP200*) inhibits autophagy in the female reproductive tract and placenta. Finally, the autophagic response varies depending on the cell type. I feel that trophoblasts sometimes show an opposite response compared with that in cancer cell lines that we generally use. For instance, the endoplasmic reticulum (ER) stress inducers, brefeldin A and tunicamycin, activate autophagy in HeLa cells [13], while these inducers block autophagy by reducing the number of lysosomes in some trophoblast-derived cells, including HchEpC1b cells, TCL1 cells, and BeWo cells [14]. Therefore, it is not surprising that trophoblast-derived cells show different autophagic responses to some chemical compounds. 

## 3. Role of Autophagy in Ovarian Functions

Systematic autophagy knockout mice have been established and show marginally light litter weight without anomalies, so-called fetal growth restriction (FGR), but not infertility or an increase in miscarriage. In the first report of systemic autophagic deletion of *Atg5* knockout mice, the pups delivered normally had to be provided milk nutrition to survive because of the defects in sucking ability [15]. Similar phenotypes were observed in *Atg7* knockout mice [16]. Considering the causes of suckling defects, restoration of *Atg5* in the central nervous system ameliorates lethality in *Atg5* knockout pups, which have normal autophagy in the brain but defective systemic autophagy [17]. *Atg5*-knockout ovaries in the pups presented normal follicular development, a lack of corpus luteum, and an increase in atretic follicles, suggesting a failure of ovulation. In addition, the uterus was defective in the development of the endometrial gland. These results indicate that genital organs require autophagy for normal development. *Becn1*-knockout mice, in which *Becn1* was conditionally deleted in granulosa cells and luteal cells, were established as ovary-specific cKO mice [18]. This mouse showed more than 75% *Becn1* reduction, accompanied by an accumulation of p62 in granulosa cells. In the reproductive phenotypes, ovulation, fertilization, and implantation were likely to be equivalent to the control; however, the conditional ablation of *Becn1* resulted in an increase in miscarriages or early deliveries due to the failure of progesterone production in the mitochondria. Pathologically, *Becn1*-knockout ovaries do not store neutral lipid droplets, the source of progesterone, in luteal cells. Although BECN1 is involved in nucleation, the first step in the autophagy pathway, multiple large autophagosomes were observed in luteal cells of the ovaries, suggesting impairment of the autophagy pathway. Although the mechanism by which *Becn1* downregulation increases the number of autophagosomes in luteal cells remains unknown, it appears that *Becn1* participates in lipid storage during pregnancy. Paradoxically, overexpression of BECN1 increases progesterone production in cultured granulosa cells, accompanied by the induction of the steroidogenic acute regulatory protein, 3β-hydroxysteroid dehydrogenase, and cytochrome P450 family 11 subfamily A member 1 (CYP11A1), enzymes that synthesize progesterone [19]. 

Follicle-stimulating hormone (FSH), which is secreted from the pituitary gland, enhances follicles from primary to dominant preovulatory follicles [20] and induces progesterone production via the BECN1 protein by degrading lipid droplets in porcine granulosa cells [21]. FSH-mediated progesterone production was diminished in porcine granulosa cells treated with chloroquine, an autophagy inhibitor, or *Atg5* siRNA, which mediates autophagosome formation, suggesting that autophagy plays a role in progesterone production in the ovaries. The significance of the transcription factor WT1 in folliculogenesis has recently been reported [22]. Overexpression of WT1, whose heterozygous mutations lead to subfertility in female mice, affects normal granulosa cell development, accompanied by the suppression of FSH receptor (FSHR) and cytochrome P450 family 19 subfamily A member 1 (also known as aromatase) expressions [23]. Inhibition of autophagy leads to the accumulation of WT1 protein in granulosa cells, which decreases the levels of cytochrome P450 family 19 subfamily A member 1 and FSHR, resulting in the disruption of granulosa cell differentiation [22]. For the regulation of WT1, *Epg5* (*ectopic P-granules 5 autophagy tethering factor*), whose deleted mice showed a phenotype similar to that of patients with premature ovarian failure, promotes WT1 degradation via p62 in granulosa cells [24]. Senescent and apoptotic cells increase with aging. WT1, which is degraded from secondary follicles to antral follicles during folliculogenesis, is sustained in the granulosa cells of the *Epg5*-knockout ovary. These changes would result in subfertility in the mice. In contrast, bone morphogenetic protein-2 enhances granulosa cell proliferation via sphingosine kinase-1 during folliculogenesis, whose differentiation is independent of FSH [25]. Since the expression of bone morphogenetic protein-2 is regulated by WT1 [23], autophagy may influence early to late folliculogenesis in the ovaries. 

Aging is an inevitable factor affecting human fertility [26]. As autophagic activity gradually diminishes in several organs, Rubicon (Run domain Beclin-1-interacting and cysteine-rich-containing protein), which suppresses autophagy by interacting with Becn1, is a central regulator of aging-inhibited autophagy [27]. In aged (24 months old) rat ovaries, the mRNA and protein levels of *Atg5*, *Atg12*, and *Becn1* are significantly lower compared to young (6 months old) rat ovaries [28]. Regarding the detailed mechanism of ovarian aging, follicular atresia, which can be induced by inflammation, mitochondrial impairment, or oxidative stress, is an important physiological change [29,30,31]. The mRNA level of *the NLR family pyrin domain containing 3* (*NLRP3*) inflammasome was significantly higher in granulosa cells from patients with diminished ovarian reserve, showing infertility [32]. Although the accumulation of p62, a marker of autophagy failure, gradually increased in an age-dependent manner in the ovary, ablation of *NLRP3* blocked the accumulation of p62 in aged ovaries [32]. Similarly, an increase in the number of atretic follicles with increased p62 accumulation has been observed in the ovaries of adult rats on a high-fat and high-sugar diet [33]. The NLRP3 inhibitor MCC950 suppressed the age-dependent increase in follicular atresia, increasing the pregnancy rate in aged mice. Autophagy plays a role in preventing the activation of inflammasomes [34]. The inhibition of inflammasomes plays a role in activating autophagy, and vice versa. 

## 4. Role of Autophagy for the Development of the Conceptus and Endometrial Decidualization

Autophagy is activated within 4 h in oocytes in response to fertilization [35]. Although many maternal proteins in oocytes are used for the synthesis of embryonal proteins, autophagy in fertilized oocytes is involved in the elimination of maternal proteins or mRNAs during the oocyte-to-embryo transition. Embryonal autophagy, which is attenuated until the middle two-cell stage, is reactivated since the late two-cell stage, autophagy-defective embryos, which are constructed by the fertilization of *Atg5*-null oocytes and *Atg5*-null sperm, stop growing at four- to eight-cell stage [35]. An aneuploid mosaic embryo, which is a major cause of miscarriage but not a normal diploid embryo, was eliminated by the increase in dying cells in the inner cell mass, generating a fetus in the peri-implantation period [36]. The increase in cell death was augmented by BAF or the genetic deletion of *Atg5*, but not rapamycin, in a p53-dependent manner [37]. Therefore, autophagy is essential for normal embryonic development in mice. After implantation, deletion of upstream factors in the autophagy pathway, such as *Becn1*, a potential mammalian counterpart of the yeast autophagy protein *FIP200*, or an activating molecule in *beclin1-regulated autophagy* (*AMBRA1*), resulted in embryonic lethality due to embryonal anomalies. In contrast, defects of downstream factors, such as *Atg5*, *Atg7*, *Atg9a*, or *Atg16L*, showed lower severe phenotypes in pups whose weight was lesser than the wild type [4]. However, pups with the deletion of *LC3B*, the most downstream factor, which is indispensable for conventional autophagosome formation, are delivered normally; their weight is similar to that of the wild type [38]. Similar results have been obtained in *GABA_A_ receptor-associated protein* (*GABARAP*) knockout mice [39]. Therefore, autophagy-related molecules can be compensated for by other molecules via a step-down process. In contrast, there are seven proteins: two splicing variants of LC3A, LC3B, LC3C, GABARAP, GABARAPL1, and GABARAPL2 [40]. The GABARAP subfamily regulates starvation-induced autophagy via ULK1 activity, whereas the LC3 subfamily negatively regulates ULK1 activity [41]. To precisely evaluate the effects of LC3s and GABARAPs on embryonic development, deletion of all subfamily genes might be required. 

Regarding other fundamental factors for implantation, except for embryos, decidualization depends on receptivity to the implant. Decidualization is associated with pregnancy and is important for functional and morphological changes in the cells of the endometrium. These changes include decidual changes in endometrial stromal cells (ESCs), recruitment of leukocytes, and morphological vascular changes. These processes are intricately regulated by the orchestration of nutrients, ovarian hormones, and cellular signaling [42]. Autophagy flux, which is evaluated by the increase in LC3-II protein in the presence of BAF or chloroquine, was increased in human immortalized ESCs with decidualization than in those without decidualization [43,44]. Decidualized ESCs recruit decidual natural killer (NK) cells, which positively regulate angiogenesis during early placentation [45] via autophagy [43]. To verify the role of autophagy in decidualization in humans, two decidualization markers, *IGFBP1* (*insulin-like growth factor binding protein 1*) and *Prolactin*, were evaluated in human ESCs, whose autophagy flux was reduced by *Atg16L1* siRNA. As a result, these markers were significantly reduced in autophagy-suppressed cells [46]. Notably, *Atg16L1* cKO in the uterus showed complete deletion of decidualization induced by sesame oil, resulting in a decrease in the implantation rate [46]. Regarding the key molecules for decidualization in the autophagy pathway, Oestreich et al. reported that *FIP200*, whose systemic deletion causes embryonic lethality due to heart failure and liver degeneration [47], was responsible for progesterone-dependent inhibition of epithelial proliferation in the murine endometrium via *Klf15* (KLF transcription factor 15) [48]. *FIP200* cKO uterus, which expressed a similar level of progesterone receptor (PR) as the control, presented complete implantation failure. Thus, *FIP200* plays a crucial role in PR signaling in the endometrium. In the cKO mice, artificially induced decidualization failed to increase the uterus’ weight. This is caused by the failure of PR-dependent permeabilization because the progesterone-PR signal induces physiological vascular permeability to promote implantation, independent of vascular endothelial growth factor [49]. In contrast, *Atg7* cKO in the uterus showed normal fertility in 8-week-old mice, which indicated no difference in the number of pups, resorption rate, fetal weight, or placental weight [50]. Although this study did not mention decidualization in the endometrium, the role of autophagy in decidualization remains controversial. If decidualization failure is partially determined by autophagy suppression, it might cause infertility and placental development. Human ESCs, which were obtained from non-pregnant women with previously severe preeclampsia, failed to decidualize adequately in vitro, accompanied by lower protein levels of insulin-like growth factor binding protein-1 and prolactin [51]. Therefore, clarifying the role of autophagy in decidualization failure could provide clues for infertility and the etiology of preeclampsia. 

## 5. Role of Autophagy in the Early Pregnant Phase to Construct Placentas 

Trophoblast stem cells differentiate from cytotrophoblasts into two cell types: syncytiotrophoblasts, which contribute to pregnancy hormone production and the barrier against pathogens, and extravillous trophoblast (EVT) cells, which anchor the placenta to the uterus and enlarge maternal vessels to obtain blood flow from the mother [4]. Syncytiotrophoblasts, formed by the fusion of cytotrophoblasts, are located on the outer surface of the villi. EVT cells migrate into the decidualized endometrium and spiral arteries. Placentas inevitably face a low-oxygen environment because of which a trophoblastic plug blocks maternal blood influx from spiral arteries during the early pregnancy period and is thought to adapt to the environment to enhance EVT invasion. The LC3 puncta, which represent the LC3-II protein, were observed in EVT cells present in the decidua at the implantation site at seven weeks of gestation [52]. Moreover, a 2% oxygen tension, which is equivalent to the physiological oxygen tension during early pregnancy, activates autophagy in primary trophoblasts [52,53]. *Low-density lipoprotein receptor-related protein 6*, whose mRNA was lower in preeclamptic placentas than in normal placentas, or the orphan nuclear receptor NUR77, which was induced in mice fed a high-fat diet, promoted EVT invasion via activation of autophagy [54,55]. Rapamycin, an autophagy activator, or high-glucose-mediated autophagy activation did not promote EVT invasion [56,57]. Although it is not easy to discuss the different results, physiological, but not pathological or excessive, activation of autophagy might be required for EVT functions. This is because autophagy-suppressed EVT cell lines constructed with stable transduction of the ATG4B^C74A^ mutant, which completely blocks LC3 conversion, showed a failure of invasion and vascular remodeling under hypoxia in vitro (Figure 3) [52]. In addition, placenta-specific *Atg7* cKO mice exhibit inhibition of placental growth, accompanied by poor invasion of trophoblasts [58]. Additionally, autophagy in trophoblasts affects normal placental development. 

Human trophoblastic syncytialization is a sequential process that requires cell growth, cell cycle arrest, and subsequent cell fusion, and this process is important for the normal functions of trophoblasts [59]. Autophagy affects syncytialization, as evidenced by increased LC3-II expression in BeWo cells following treatment with forskolin, a cAMP activator [60,61]. Meanwhile, LC3-II was increased in syncytialized cells, but the increase appeared to be saturated in terminally syncytialized trophoblasts in primary human trophoblasts [62]. Downregulation of the transcription factor EB (TFEB), a basic helix-loop-helix-zipper family member and a central regulator of autophagy and lysosomal biogenesis [63], is accompanied by p62 accumulation in syncytial BeWo cells (Figure 3), supporting the observation that autophagy is suppressed during syncytialization. *TFEB-*deficient placentas failed to construct the labyrinth layer owing to vascular endothelial growth factor defects, resulting in embryonic lethality [64]. Since the *Atg7*-cKO placentas, in which TFEB was downregulated, also showed a reduction in *placental growth factor* mRNA levels [58], TFEB might be involved in villous construction via autophagy. In addition, BAF inhibited syncytial fusion and human chorionic gonadotropin (hCG) production in BeWo cells and primary human trophoblasts (Figure 3) [62]. Although the effects of angiogenic factors have not yet been demonstrated, it would be noteworthy to determine whether BAF affects the secretion of *placental growth factor* and sFLT1 (soluble fms-like tyrosine kinase 1) in placental tissues. 

Mitochondrial energy production is centrally involved in syncytialization [65]. AMP-activated protein kinase (AMPK) stimulation activates mitochondrial autophagy, mitophagy, in BeWo cells [66], and the knockdown of *AMP-activated protein kinase* increases the mitochondrial volume [67], suggesting that autophagy maintains the quality of mitochondria in syncytiotrophoblasts. A reduction in mitofusin 2 (MFN2) has been reported in cases of unexplained miscarriages [68]. As *MFN2* deficiency causes the accumulation of damaged mitochondria by the failure of autophagy and impaired metabolic homeostasis by reactive oxygen species, the downregulation of MFN2 might be related to a defect in syncytialization, which leads to miscarriage. As anti-phospholipid antibodies are an increased risk for recurrent pregnancy loss in pregnant women, one of the antibodies, anti-β2-glycoprotein I antibodies, has a function to activate inflammasomes in the human first-trimester trophoblasts [69]. Autophagy, an inflammasome inhibitor [70], prevents inflammasome formation in response to ER stress in trophoblasts [71] and is involved in the protection against recurrent pregnancy loss due to the formation of antiphospholipid antibodies, which enhance inflammasome activity [69]. 

## 6. Role of Autophagy during the Middle to Late Phase to Maintain Pregnancy

To our knowledge, syncytiotrophoblasts are large multiple-nucleic cells that cover the surface of the microvilli facing the maternal blood. Pathological changes in syncytiotrophoblasts of preeclamptic placentas are characterized by accelerated maturation, fibrin deposition, and syncytial knots, which are often observed during the early onset of preeclampsia. Redman et al. proposed the following 12 syncytiotrophoblast stress lesions of preeclampsia: syncytial clubbing and necrosis, syncytial knots, mitochondrial stress, apoptosis, complement deposition, senescence, oxidative stress, ER stress, inflammatory stress, autophagy, pyroptosis, and perivillous fibrin deposition [72]. In addition, the acceleration of placental aging in early-onset preeclampsia, estimated by DNA methylation at 62 CpG sites, has been the focus of the research [73]. Although higher maternal age increases the incidence of preeclampsia in pregnant women, autophagic inhibition may accelerate placental aging in preeclamptic placentas because autophagic activity gradually decreases with age [27]. Since autophagy is involved in sustaining cellular homeostasis, evidence has accumulated for autophagic protection against stress in the placenta. 

There are conflicting reports on LC3-II and p62 expression, the major markers of autophagic status in autophagy-related proteins in placental tissues [9,74]. As explained in Section 2, “Things that you have to know for autophagic experiments”, it is difficult to speculate autophagy status only by LC3-II and p62 expressions in fixed human placental tissues. Declines in other autophagy-related proteins, such as ATG3, ATG101, ATG5-12 complex, LAMP1, LAMP2, and TFEB, have been reported, indicating the inhibition of autophagy in preeclamptic placentas [9,75,76]. Excessive autophagic activation, mediated by ceramide, CYP11A1, or the inhibition of protein kinase C isoform β, has been reported in preeclamptic placentas [77,78,79,80,81]. Both directed autophagic dysregulations would contribute to the pathophysiology of preeclampsia because both directions indicate a homeostatic disruption in trophoblasts. Therefore, dysregulated autophagy might be linked to phenotypic differences in preeclampsia, such as early or late onset, the severity of hypertension, renal dysfunction, and FGR complications. Although poor placentation is a common pathological feature between FGR and preeclampsia with FGR, the expression levels of p62 and LC3-II differ depending on the phenotype [77]. Further complex studies are required to clarify the role of autophagy in preeclampsia or FGR. 

The hypoxia-reoxygenation (H/R) model is well used for in vitro preeclamptic placental models because H/R poses ER and oxidative stress in trophoblasts [82,83]. Prolonged hypoxia also induces ER stress, resulting in an increase in pyroptosis, an inflammasome-associated cell death, which is enhanced in autophagy-deficient trophoblast cells [71]. Lipopolysaccharide-mediated inflammasome activation, which requires TANK-binding kinase 1 (TBK1) in trophoblasts, was ameliorated by the autophagy activators, rapamycin or torin1 [84]. The lack of TBK1 fails to recruit the nuclear dot protein 52 kDa, a selective autophagy receptor, to autophagosomes capturing intracellular bacteria (also known as xenophagy) [85]. Thus, autophagy protects against inflammasome induction in trophoblasts, and TBK1 may be involved in the correlation between autophagy and inflammation during pregnancy. Regarding the effect of syncytiotrophoblastic stress on autophagy, ER stress inducers inhibited autolysosome formation, which is structured by the fusion of autophagosomes and lysosomes, due to the decrease in lysosomes. In contrast, autophagy inhibitors increased the expression of BIP (binding immunoglobulin protein, also known as HSPA5), an ER stress marker protein, in trophoblasts, as shown in the supplemental Figure [14]. H/R treatment critically decreases the numbers of autophagosomes and autolysosomes in trophoblasts, resulting in an increase in aggregated proteins in the cytoplasm due to the failure of autophagy [75]. Thus, ER stress enhances autophagy inhibition, and vice versa (Figure 4).

What are the effects of oxidative stress on autophagy? Hydrogen peroxide activated autophagy in some trophoblast cell lines in a dose-dependent manner, accompanied by an increase in phosphorylated-p62 (p-p62), a marker of protein aggregation [86]. Accumulation of p-p62 induces heme oxygenase-1 (HO-1), an antioxidant enzyme, by dispersing Keap1-Nrf2 conjugation [87], whereas BAF, an autophagy inhibitor, significantly decreases the expression of HO-1 and NADPH quinone dehydrogenase 1, another antioxidant enzyme, in BeWo cells, accompanied by a higher accumulation of p-p62 (Figure 4). To answer this question, reports stated that the neighbor of BRCA1 gene 1 (NBR1), an autophagy receptor-like p62, is required to induce NADPH quinone dehydrogenase-1 via Nrf2 isolation in response to oxidative stress [88]. An increase in the p-p62 protein level following BAF treatment is accompanied by a significant decrease in NBR1 expression in BeWo cells [86]. Intriguingly, chloroquine, another autophagy inhibitor used in pregnant women with systemic lupus erythematosus, did not affect HO-1 or NBR1 expression. As chloroquine also has different effects on autophagic inhibition, compared with BAF [7], it would be safer in pregnant women. Although hydrogen peroxide ameliorates BIP expression in BeWo cells in a dose-dependent manner (unpublished data), it might be dependent on autophagy activation because autophagy protects against environmental oxidative stress [89]. Thus, the protective function of autophagy in response to oxidative stress might rely on NBR1 because NBR1 is essentially involved in the oxidative stress response but not in the macroautophagy machinery [88]. In summary, oxidative stress can activate macroautophagy in trophoblasts, but the p62-NBR1-HO-1 axis might respond to oxidative stress independent of macroautophagy. Excessive ER stress inhibits macroautophagy. If ER stress or other stimuli inhibit NBR1 expression, the viability of trophoblasts might be compromised due to excessive oxidative stress. 

## 7. Conclusions

Thus far, the role of autophagy in reproduction has been examined, analyzed, and discussed based on the idea of macroautophagy. However, among the autophagy machinery, microautophagy and chaperone-mediated autophagy share lysosomes with macroautophagy for the degradation of their targets but not autophagosomes [90]. In addition, LC3-mediated non-canonical autophagy has been discovered [91,92]. Autophagy fundamentally maintains the quality of organelles in cells by degrading mitochondria, lysosomes, or the ER, and by reproducing lysosomes [93]. Therefore, autophagy is a complex process that plays a role in maintaining cellular homeostasis; however, further studies are required for a better understanding. Whether inhibition or activation of autophagy contributes to the occurrence of preeclampsia is still controversial, we believe that the maintenance of normal autophagy is necessary for reproductive health. 

## Figures and Tables

**Figure 1 biology-12-00373-f001:**
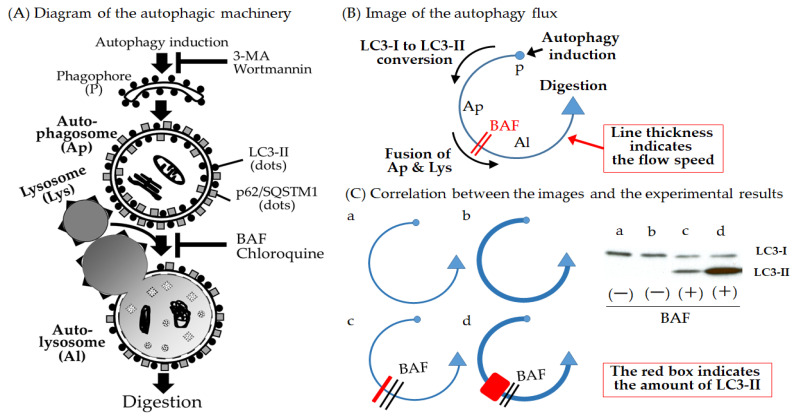
Diagrammatic autophagy. (**A**) Diagram of the autophagic machinery. (**B**) Three-quarter circle image of the autophagy flux. Autophagy constantly flows in the cells. The line thickness indicates the flow speed. (**C**) The imaged experimental results. The three-quarter circle images indicate “Low autophagic flow” (a, c) and “High autophagic flow” (b, d). The indicated Western blotting image shows an ideal result. The image was obtained from HchEpC1b cells, a trophoblast-derived cell line, which were cultured in DMEM/10% FBS (a, c) or DMEM/0% FBS (b, d) with (c, d) or without (a, b) BAF. As LC3-II, but not LC3-I can be visualized with BAF, the amount of LC3-II (red boxes) ± lysosomal inhibitors indicates the autophagy flux. P: phagophore; Ap: autophagosome; Lys: lysosomes; Al: autolysosomes. BAF: bafilomycin A1.

**Figure 2 biology-12-00373-f002:**
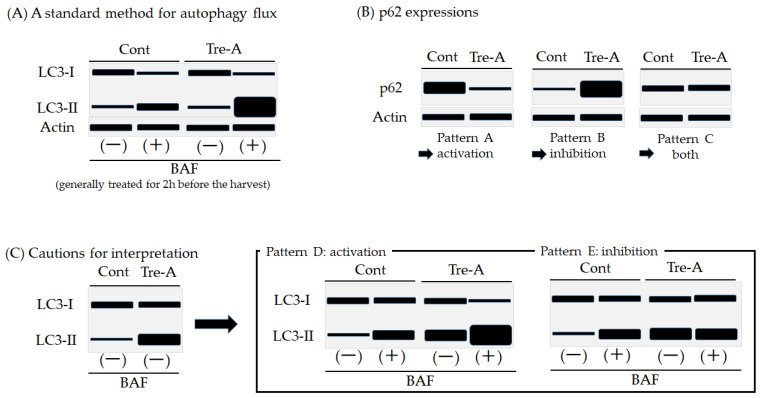
The methods for evaluating autophagy flux. (**A**) Treatment A (Tre-A) increases LC3-II protein in the presence of BAF. The ratio of LC3-II/Actin with BAF is higher in the Tre-A than control (Cont), indicating autophagy activation. (**B**) Examples of p62 expression with some treatment. Pattern A is a typical result that indicates autophagy activation by the treatment. Pattern B highly suggests autophagy inhibition because an autophagy receptor, p62, is accumulated. Pattern C can be seen in both statuses. (**C**) This result does not indicate autophagy activation by Tre-A. When autophagy flux is seen with BAF, autophagy is activated (pattern D). Meanwhile, LC3-II does not increase in the cells with Tre-A in the presence of BAF, indicating autophagy inhibition (pattern E).

**Figure 3 biology-12-00373-f003:**
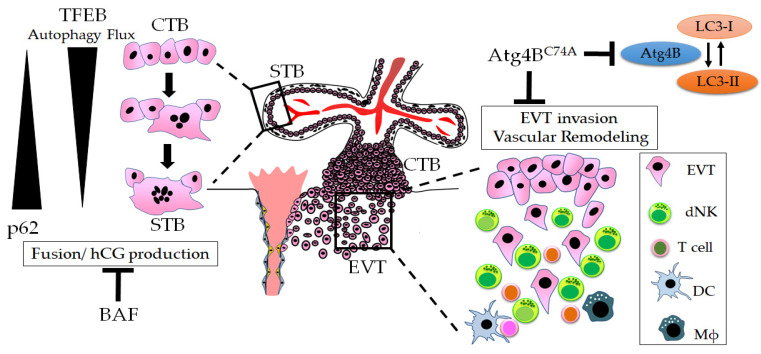
The roles of autophagy in early placentation. Autophagic activity is decreased during syncytialization in the in vitro model. BAF inhibited syncytial fusion and human chorionic gonadotropin (hCG) production in STB. EVT cells invade into decidualized endometrium under hypoxia, and face various types of maternal immune cells, such as decidual NK (dNK) cells, T cells, dendritic cells (DC), or macrophages (MΦ). Invasion and vascular remodeling are inhibited in EVT cells transduced with the Atg4B^C74A^ mutant. STB, syncytiotrophoblasts; CTB, cytotrophoblasts; EVT, extravillous trophoblasts.

**Figure 4 biology-12-00373-f004:**
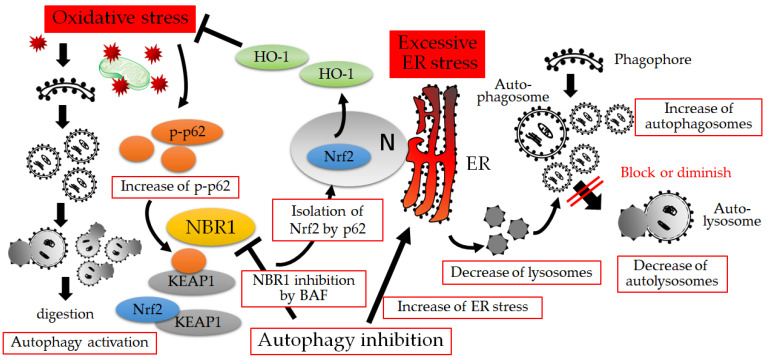
Oxidative stress (OS) and endoplasmic reticulum (ER) stress on autophagy. OS activates autophagy in trophoblasts. The induction of HO-1, an anti-OS enzyme, is dependent on the p62-NBR1-Nrf2 axis, but not the general autophagy pathway. Excessive ER stress inhibits autophagy via the decrease of lysosomes, resulting in the increase of autophagosomes. Bafilomycin (BAF) suppresses NBR1 expression, suggesting that autophagy inhibition by ER stress attenuates the anti-OS response. HO-1: heme oxygenase-1; KEAP1: Kelch-like ECH-associated protein 1; N: nucleus; Nrf2: nuclear factor-erythroid 2-related factor 2; p-p62: phosphorylated p62.

## Data Availability

Not applicable.

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
