# Peer review of "The Role of Autophagy in the Female Reproduction System: For Beginners to Experts in This Field"

_biology, 2023, doi:10.3390/biology12030373_

Round 1

Reviewer 1 Report

The authors reviewed roles of autophagy in female reproductive health.  They first introduced basics of autophagy, including the commonly used experimental methods, then intended to update readers with current status of autophagy research in female gamete and embryos, particularly around the time of implantation.  The information provided in the manuscript should be interesting to researchers in cell and reproductive biology.  However, several issues may need to be addressed for the manuscript before publication, including:

1.     A general introduction of autophagy with the known key regulators’ functions and types of autophagy should be provided in the beginning (e.g. what ATG proteins do), so that readers could follow in the successive sections.

2.     Many ambiguous terms are used in the text, which could be replaced by more specific descriptions, such as: homeostasis (tissue, cellular or protein?), this protein (line-32), this increase (line 42), high barriers (any specific issues?) (line-510, intact development (line (262), forskolin is for PKA (line265), etc.

3.     It will be helpful if the working mechanisms of inhibitors (e.g. BAF, chloroquine), LC3-I/II and p62 could be briefly introduced before addressing their usage (lines 65-66, lines 68-70, lines 81-83).

4.     Would it better to understand the “opposite effect” the authors mentioned as the “cell-type specific effect” of autophagy (lines 105-108)?

5.     Some specific examples may be provided for the first mentioned cKO and KO mouse models (lines 104, 111), for readers to know what the authors mean.

6.     Lines 145-147 and lines 150-153 seem repeating.

7.     A figure (summary or model) would be helpful to better follow Section 3.

8.     A brief introduction of “decidualization” at the beginning of Section 4 would be helpful.

9.     Real Western blots should be used in Figure 2 when discuss the experiments.

10.  The manuscript should be carefully checked and edited for language, including phrases and grammar errors.

11.  Since the manuscript mainly discusses research in female reproductive system, an alternative title may be considered.

12.  The resolution of all figures need to be improved. 

Reviewer 2 Report

In this manuscript, the authors summarize studies on the role of autophagy in reproduction in different models, with an introduction on how to interpret LC3 Western blots.

The review is timely and interesting, however some comments need to be addressed before publication.

In section 2 the authors conclude that autophagy assessment in fixed tissue is hard to interpret. Please comment on possible ways to overcome this difficulty. Some suggestions have been suggested in: Humbert, M.; Morán, M.; de la Cruz-Ojeda, P.; Muntané, J.; Wiedmer, T.; Apostolova, N.; McKenna, S.L.; Velasco, G.; Balduini, W.; Eckhart, L.; Janji, B.; Sampaio-Marques, B.; Ludovico, P.; Žerovnik, E.; Langer, R.; Perren, A.; Engedal, N.; Tschan, M.P. Assessing Autophagy in Archived Tissue or How to Capture Autophagic Flux from a Tissue Snapshot. Biology 20209, 59

My minor comments:

Line 32, it says This protein when referring to autophagy. Autophagy is not a protein, it is a process, please correct.

Line 37, it says degeneration, please replace with degradation

Figure 1B, it says: Please indicate the Western Blot conditions in the figure legend. Also in the legend, please change: the amount of LC3-II (red boxes) indicates the autophagy flux to: the amount of LC3-II (red boxes) +/- lysosomal inhibitors indicates the autophagic flux.

Line 200-202, the authors mention that pups with the deletion of LC3B are delivered normally in comparison to ATG gene knockout mice. Several members of the LC3/GABARAP family have been described that could compensate in this fenotype (described in Schaaf MB, Keulers TG, Vooijs MA, Rouschop KM. LC3/GABARAP family proteins: autophagy-(un)related functions. FASEB J. 2016 Dec;30(12):3961-3978). Please discuss.

Lines 288-293. It says:        Autophagy, which has an 290 inhibitory function on inflammasome activation [61], prevents inflammasome generation 291 induced by endoplasmic reticulum (ER) stress in the trophoblasts [62] and is involved in 292 recurrent pregnancy loss induced by anti-phospholipid antibodies. If authophagy inhibits inflammasome activation, would it be involved in recurrent pregnancy loss or would it prevent pregnancy loss? Please clarify.

Figure 3, please define, in the legend, the immune cells presented in the figure.

Line 354. It says, an increase in p-p62 expression by BAF. Do the authors mean gene expression? If they are referring to protein levels, it should say p-p62 protein levels.

Figure 4. It says, in the figure, NBR1 inhibition by BAF not CHQ. This would mean that autophagy inhibition is not responsible for the observed phenotype and then should not be within the autophagy inhibition arm. Please explain. Also, for Figure 4, define all the abbreviations in the figure legend.

Reviewer 3 Report

Nakashima and collaborators wrote a very interesting and well-structured article. Moreover, this manuscript will be meaningful to the autophagic reproductive field article. However, the paper has some minor errors that should be addressed before publication.

Most of the paragraphs have good fluency and are easy to read. However, many of them need the proper references. Please, add carefully throughout the text.  

The authors wrote Atg7, Atg5, etc., incorrectly throughout the text. For example, in line 117, “specific restoration of Atg5 in the central nervous system”. Authors must consider that the meaning of Atg is Autophagic-related, but it doesn’t indicate gene or protein. The authors are referred to the following article.

Klionsky D. J. (2012). Look people, "Atg" is an abbreviation for "autophagy-related." That's it. Autophagy8(9), 1281–1282. https://doi.org/10.4161/auto.21812

Line 32 states, “This protein is conserved from yeast to mammals.” The authors must change “This protein” to “This process” because the authors are describing the autophagic process.

In line 34, the authors should add that the autophagic process is always active at low basal levels.

Authors are encouraged to discuss further why LC3-II/LC3-I ratio is not recommended to evaluate autophagy flux. In some tissues, the levels of LC3-I are cell, stress, and inducer-dependent. The authors are referred to Guidelines for the use and interpretation of autophagy, section 2a. Atg8-family protein detection and quantification pages 53-59

Klionsky, D. J., Abdel-Aziz, A. K., Abdelfatah, S., Abdellatif, M., Abdoli, A., Abel, S., Abeliovich, H., Abildgaard, M. H., Abudu, Y. P., Acevedo-Arozena, A., Adamopoulos, I. E., Adeli, K., Adolph, T. E., Adornetto, A., Aflaki, E., Agam, G., Agarwal, A., Aggarwal, B. B., Agnello, M., Agostinis, P., … Tong, C. K. (2021). Guidelines for the use and interpretation of assays for monitoring autophagy (4th edition)1Autophagy17(1), 1–382. https://doi.org/10.1080/15548627.2020.1797280

In line 103, the authors are encouraged to indicate the ATG gene knockout.

In lines 105-108, please briefly discuss the differences in the responses among trophoblasts and cancer cell lines. Authors are encouraged to list which are the chemical compounds.

Reviewer 4 Report

The paper looks well structured, organized, articulated and informative. The diagrams clearly explain the information in the particular sections and are easy to understand for the reader. Moreover, section 2: 'Things that you have to know for autophagic experiments' is appreciative for the beginners in autophagy. However, I do have some suggestions for improving the article if followed which are:

  1. The title of the review paper would be more appropriate and focused if it is considered/ related to 'female reproduction and development'. Since there is nothing for male reproduction in this paper, albeit it is well reviewed by Zhu et al., 2019. Autophagy in male reproduction [PMID: 31014114]. 
  2. in section 2; [line 54] Things that you have to know for autophagic experiments: It would be wonderful if you could elaborate on why it is important to check autophagy flux before experiments, as well as various factors that influence autophagy flux levels.
  3. As you mentioned about ageing and autophagy [line 163], I believe an association of metabolism and stress on autophagy process as an example could be a good fit to delineate as well. You may count on the reference:

 Rejani et al., (2022). High fat-high fructose diet elicits hypogonadotropism culminating in autophagy-mediated defective differentiation of ovarian follicles. [PMID: 36359843].

4    4Line 137, Of course StAR and CYP11A1 contribute to entry of cholesterol (lipid) into the mitochondria and conversion into pregnenolone, which in turn converted into P4 by 3β-HSD. Did the referenced author check the status of 3β-HSD? I think it should be mentioned.

55.      There is a dull conclusion at the end of the paper. Having a future perspective would strengthen it in my opinion. 

Round 2

Reviewer 1 Report

The manuscript has been improved, including language and the resolution of figures.  

The main concern is the Figure 2, which uses schematic drawings to describe experimental results, even if they are predicted possibilities.  Since potential outcomes and assays are built upon experiments or experiences that may have been reported or done previously, real Western blots should be used as examples, in order to demonstrate the situations, for example, Treatment A (a real example of the treatment) and to avoid potential misleading for readers.  Some previous published results and references could be included for Figure 2.
